# How do host population dynamics impact Lyme disease risk dynamics in theoretical models?

**Joseph D. T. Savage**[1,2]*, **Christopher M. Moore**[1]

**1** Biology Department, Colby College, Waterville, Maine, United States of America, **2** Department of Geography, Graduate Program in Ecology, Evolution, Environment, and Society, Dartmouth College, Hanover, New Hampshire, United States of America

* joseph.savage.3927@gmail.com

**Data Availability Statement:** Inputs and test output data can be found on Dryad at: https://doi.org/10.5061/dryad.pc866t1xt. Model code can be

## Abstract

Lyme disease is the most common wildlife-to-human transmitted disease reported in North America. The study of this disease requires an understanding of the ecology of the complex communities of ticks and host species involved in harboring and transmitting this disease. Much of the ecology of this system is well understood, such as the life cycle of ticks, and how hosts are able to support tick populations and serve as disease reservoirs, but there is much to be explored about how the population dynamics of different host species and communities impact disease risk to humans. In this study, we construct a stage-structured, empirically-informed model with host dynamics to investigate how host population dynamics can affect disease risk to humans. The model describes a tick population and a simplified community of three host species, where primary nymph host populations are made to fluctuate on an annual basis, as commonly observed in host populations. We tested the model under different environmental conditions to examine the effect of environment on the interactions of host dynamics and disease risk. Results show that allowing for host dynamics in the model reduces mean nymphal infection prevalence and increases the maximum annual prevalence of nymphal infection and the density of infected nymphs. Effects of host dynamics on disease measures of nymphal infection prevalence were nonlinear and patterns in the effect of dynamics on amplitude in nymphal infection prevalence varied across environmental conditions. These results highlight the importance of further study of the effect of community dynamics on disease risk. This will involve the construction of further theoretical models and collection of robust field data to inform these models. With a more complete understanding of disease dynamics we can begin to better determine how to predict and manage disease risk using these models.

## Introduction

Emerging infectious diseases present an ever-growing threat to human health [1–5], agriculture [6], and native flora and fauna [2]. A growing proportion of new emerging diseases in humans are vector-borne, and diseases spread by vector organisms are the most common

found on GitHub at: https://github.com/JDTSavage/Lyme_Host_Dynamics.

**Funding:** JS, Buck Lab for Climate and Environment at Colby College: https://web.colby.edu/bucklab/ JS, Student Special Project Fund: https://www.colby.edu/academics/departments-and-programs/environmental-studies/opportunities/student-research-grants/ JS, Colby Summer Research Assistantship: https://web.colby.edu/linde-packman-lab/colby-summer-research-program/ JS: National Science Foundation (OIA 2019609). https://www.nsf.gov The funders had no role in study design, data collection and analysis, decision to publish, or preparation of the manuscript.

**Competing interests:** The authors have declared that no competing interests exist.

across pathogens and hosts [1, 4–6]. These disease systems involve complex dynamics between wildlife hosts, vector organisms, and the disease itself. The study of vector-borne diseases is of interest to many fields, including ecology, epidemiology, climatology, and many other disciplines [2, 4, 5]. In humans, around 60% of emerging diseases are wildlife zoonoses, over 70% of which originate in wildlife [5]. Lyme disease is the most prominent emerging vector-borne zoonosis impacting human health in the United States, with estimates of as many as 476,000 cases per year [7].

A key problem in disease ecology is understanding how to predict and control the proliferation of vector-borne diseases [8, 9]. As hosts and pathogens, the structure and dynamics of these diseases are best thought of as communities or networks of interacting species [2, 10]. Studying the community ecology of these disease systems is fundamental to understanding how they develop and spread. Much research has focused on "simpler" applications of community and population ecology, such as single host-parasite interactions. The tools of community ecology allow an understanding of the mechanisms leading to the particular community assemblages we see today and the dynamics that shape the systems across time and space, including the emergence and spread of vector-borne diseases [4, 5]. This has particular benefit in informing how host communities and parasite communities develop and interact between levels and different scales, from within the individual to across ecosystems.

In studies on vector-borne diseases, research has aimed to understand the impact of the traits (e.g. reservoir competency, mean population density) of different species and species communities on disease risk [10–20]. Lyme disease exemplifies this research. The traits of host species, such as reservoir competency and the number and life-stage of ticks that an individual host can support are well known. However, the impact of host and community dynamics on Lyme disease risk is fairly uncertain considering the widely implicated potential of these dynamics to have a significant impact on how *Borrelia burgdorferi*, the pathogen responsible for Lyme disease in humans, flows through wildlife communities. Rodents are known to serve as important reservoir hosts for *B. burgdorferi* [10, 16, 21, 22], and populations of these species often undergo extreme fluctuations in density driven by resources [10, 20, 21], predators [10, 22], and environmental effects [10, 23]. Other host species undergo population dynamics and many species are subject to changing populations and extinctions as a result of human caused disturbances [10, 23]. Research has been equivocal in determining the effect of host dynamics on disease, showing no effect or an inconsistent effect [24–27]. Understanding this process will allow us to better understand how different systems experience disease risk, and inform how disease may vary regionally or as species extinctions and invasions develop.

Disease models of Lyme disease risk treat populations of tick hosts as static parameter values, with only a few models investigating the impact of using dynamic population data [26, 28–31], and to our knowledge, no theoretical study has investigated the impact that host dynamics have on this system. It is unclear if dynamic host populations might promote or reduce disease risk, and what the impacts might be on temporal disease dynamics. To this end, we have created a dynamic population model of *Borrelia burgdorferi* prevalence to investigate how hosts dynamics in small mammal hosts can impact model predictions of disease risk.

To investigate the impact of host dynamics on Lyme disease risk, we have created a mathematical model that simulates the flow of pathogen between a stage-structured compartment model of ticks and a small community of tick hosts. In the model, densities of the primary host, mice, are varied on an annual basis, as commonly observed in natural mouse populations [10, 20–22]. By changing the mean density and variation from the mean density in a collection of simulations, we will use our model to explore the impact that simple host dynamics have on disease risk measures of the model. Greater or lesser variation from the mean mouse density in a simulation will serve to represent potential variability within host populations, and test the

impact of a range of population dynamics. We will run these simulations under different environmental conditions to determine the robustness of the impact from host dynamics under different conditions. This model aims to direct future research into the effect of host dynamics and disease risk and highlight the need for the further development of models and empirical studies investigating this topic.

## Materials and methods

### Tick population structure

The model consists of a collection of empirically-informed discrete-time difference equations representing the populations of a tick species and three hosts (S1 File). Equation parameters are listed in S4 and S5 Tables. The tick population is stage-structured and details each of the main life stages and substages through which an *Ixodid* tick will evolve develop on a weekly time scale (Fig 1, S2 Table).

The tick population is highly detailed to ensure realistic dynamics in vector abundance and disease transmission. Ticks that enter a stage in a given week are tracked as a cohort and undergo the appropriate processes for that stage (Fig 1, S3 Table). Ticks of a stage in one week are a function of those that have survived from the previous week, with a gain or loss of density in one stage from development, survival, host finding, and infection of susceptible ticks.

The maximum time which a tick may spend in a life stage is unclear from laboratory [32, 33] and field studies [34–36], and models have incorporated a variety of biologically-informed assumptions to give life stage limits [26, 28, 29, 31]. We have set 52 weeks as a maximum limit that ticks of a cohort are tracked. We used cumulative degree weeks (CDW) to determine development rates between major life stages (egg, larvae, nymph, adult) as CDW has been shown to relate well to tick phenology and is a standard in many models [26, 29, 37, 38]. The cuticle hardening period undergone by free-living stages is a final point of confusion, with estimates ranging from 0–4 weeks, and some models only considering hardening in larvae. We have chosen a hardening period of one week for all free-living stages.

### Host population structure

There are three hosts in the model: mice, a medium-sized mammal specieshost (e.g., skunks, raccoons, foxes), and deer. Medium mammals and deer remain at constant densities of 4 and 0.4 ind. ha$^{-1}$, respectively. These values were selected from the literature to keep tick densities within that predicted by previous models and aide in potential comparisons [26]. Both species have a constant survival rate to provide lifespans of 2 and 3 years, respectively. This rate allows infection to "clear" from the host population as offspring are assumed to be born susceptible to infection. Mice have a clearing rate for a 1 year lifespan. Lifespans were estimated from literature for mice [39–41] and deer [42], and 2 years was chosen for medium mammals to provide a suitable intermediate sized host.

At the beginning of each year ($y$), mouse densities ($M$, Eq 1) are drawn from a normal distribution ($\mathcal{N}$) to simulate the near-random population dynamics induced by seed masting [21]. Density values are bounded by zero, and by varying the mean ($\mu$) and variance ($\sigma^2$) of the distribution a range of population dynamics can be simulated, including $\sigma^2 = 0$, or constant mouse density (Fig 2). For an increase in mice, the difference ($\pm\Delta_M$, Eq 2) is added to the population of susceptible ($S$) individuals (Eq 3), while a decrease proportionately affects both

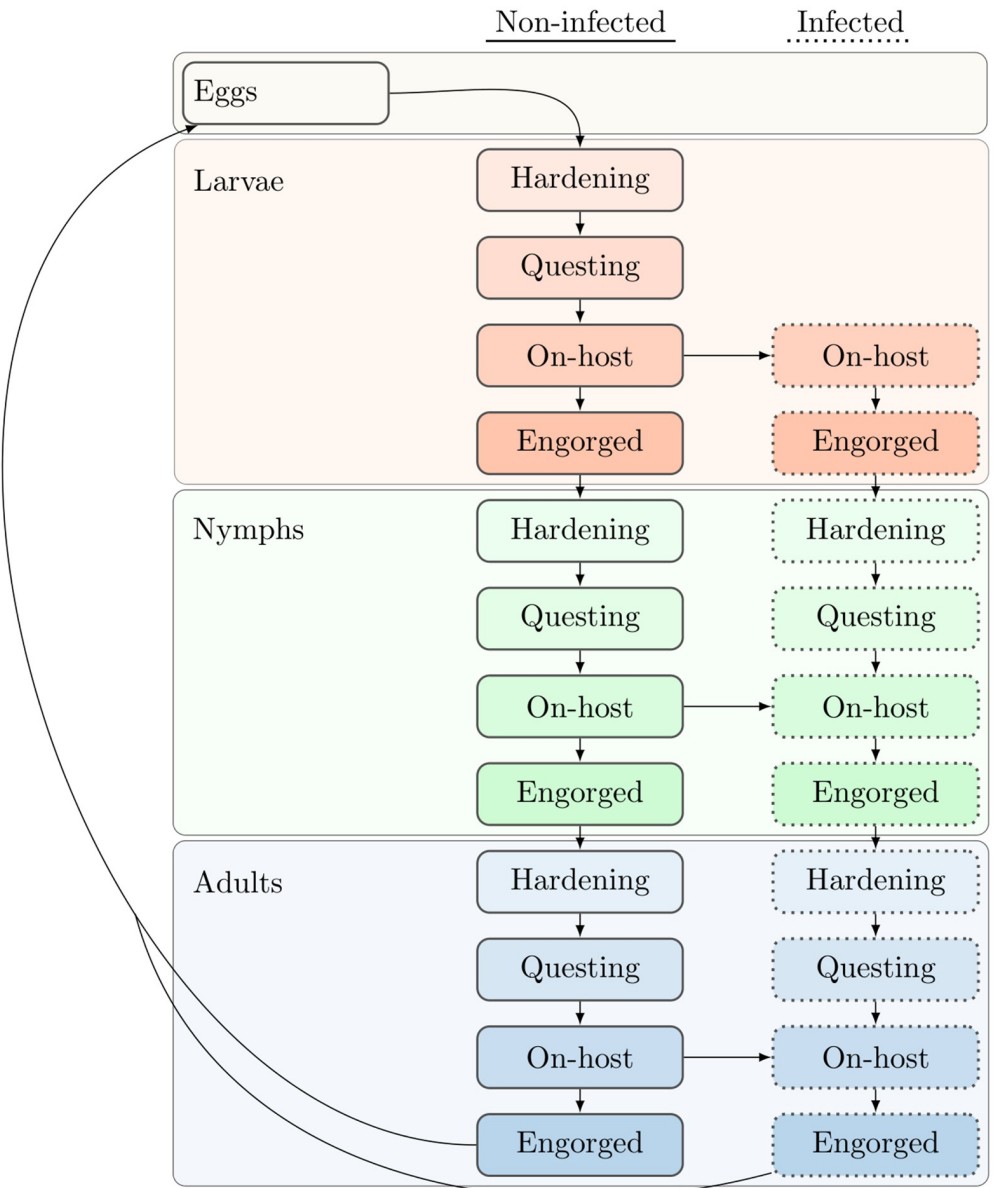

**Fig 1. Flow of ticks through different developmental stages and infection.** The structure and flow of the tick population through different developmental stages is shown. Infected ticks go through the same stages but are kept as a separate population. Infection occurs during blood meals on hosts, which can transmit to and from ticks and their hosts. New eggs from infected and uninfected adults produce the next generation of uninfected ticks.

susceptible and infected ($I$) individuals (Eq 4).

$$M_{y+1} = \mathcal{N}(\mu, \sigma^2) \tag{1}$$

$$\Delta_M = M_{y+1} - (S_y + I_y) \tag{2}$$

$$S_{y+1} = S_y + \Delta_M, \quad \Delta_{\mathbf{M}} > \mathbf{0} \tag{3}$$

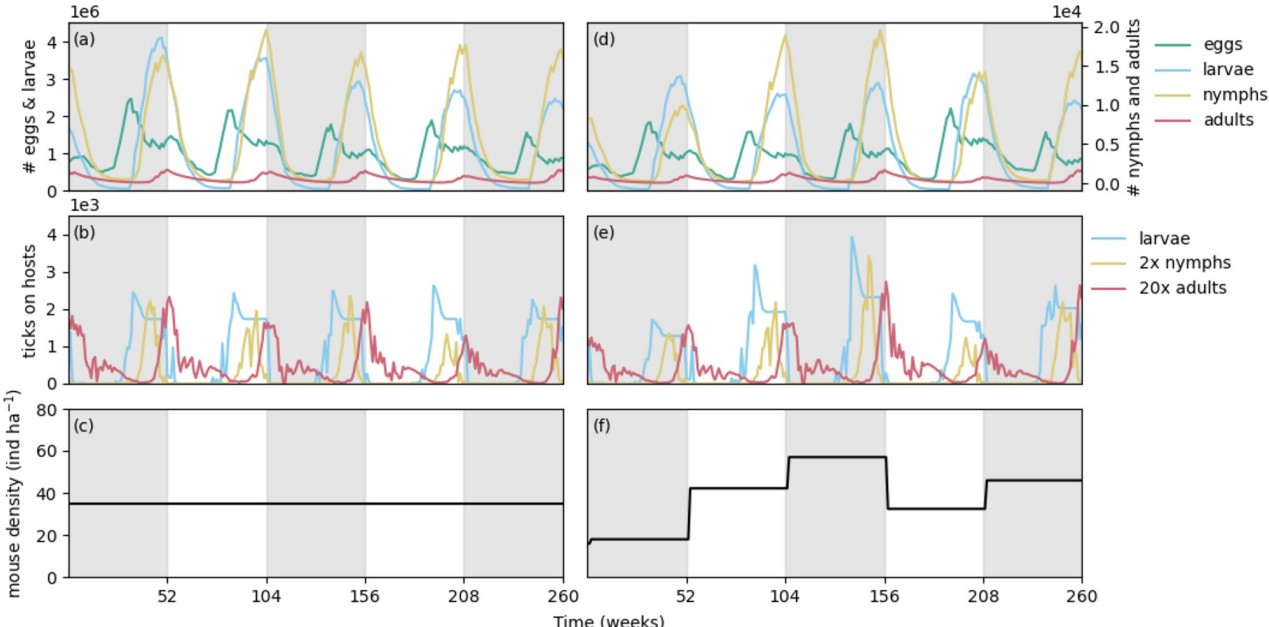

**Fig 2. Sample time series of model output with or without mouse population dynamics.** Data from two simulation runs are shown in the left and right columns, with a mean density of mice $\mu = 30$ individuals per hectare and no population dynamics ($\sigma^2 = 0$) or a standard deviation of 25 individuals per hectare around this same mean (host dynamics). All other input and parameters remain the same. Shading indicates each segment of 52 weeks or one year. (a) and (d) show the total density of eggs and questing larvae, nymphs, and adults. As is apparent, the questing populations of nymphs and adults are significantly smaller than the total population of questing larvae, as most larvae do not manage to survive and find hosts. Populations fluctuate on an annual basis but with different phases. In (d) differences in density are apparent as a result of variable host populations. (b) and (e) show the populations of on host larvae, nymphs, and adults over time. Again, these densities cycle yearly, and changes in on host density are apparent with $\sigma^2 = 25$ in (e). (c) and (f) demonstrate the difference in mouse density for a simulation without dynamics (c) and with dynamics (f) in mouse density. At the beginning of each year the population of mice is changed via a random distribution.

$$S_{y+1} = S_y + \Delta_M \frac{S_y}{S_y + I_y}, \quad I_{y+1} = I_y + \Delta_M \frac{I_y}{S_y + I_y}, \quad \Delta_{\mathbf{M<0}} \tag{4}$$

## Activity and host finding

Questing ticks seek out hosts each week. The total rate at which hosts are found is determined by the host finding rate and the tick activity rate. The host finding rate ($F_x$) for a host species ($x$) is modeled as in other discrete-time tick models [26, 29] as this strategy yielded realistic rates of host finding for a sustained tick population. In 11 Eq 5, $a$ is a host species and tick life stage ($i$) dependent coefficient, which relates the base finding rate of a tick to host density ($H_x$). The exponent of 0.515 scales down the rate of host finding to produce realistic densities [29]. The host finding rate is scaled via the level of tick activity, with $F_x$ being the maximum host finding rate assuming 100% tick activity. Tick activity ($A$, Eq 6.) is calculated with a normal distribution ($\mathcal{N}$) centered on an optimal activity temperature ($t_{opt}$), with a variance ($\sigma^2$), both determined from empirical measurements of activity levels across temperature [43, 44]. The final number of questing ticks ($Q_x$, Eq 7.) which find a host $x$ in a given week is the product of these rates and the total questing density. Each host species has a maximum host burden. If new questing ticks will cause hosts to exceed their burden in the successive week, ticks which

exceed this threshold will remain in the questing population.

$$F_x = a_{i,x}(H_x)^{0.515} \tag{5}$$

$$A = \mathcal{N}(t_{opt}, \sigma^2) \tag{6}$$

$$Q_x = F_x A Q \tag{7}$$

## Tick survival

Eggs, questing, and engorged ticks exhibit environmentally dependent survival ($S_e$), modeled with normal distributions ($\mathcal{N}$) for temperature ($T_s$, Eq 8.) and precipitation index ($P_s$, Eq 9.) dependent survival. Optimal temperature ($t_s$) and precipitation index ($p_s$) values were chosen from the literature and the variance of distributions were adjusted to produce reasonable tick densities [26, 29, 34, 36]. This eliminates assumptions made in previous models, as exact relationships between environmental conditions and survival have not been determined.

$$T_s = \mathcal{N}(t_s, \sigma^2), \quad P_s = \mathcal{N}(p_s, \sigma^2) \tag{8}$$

$$S_e = T_s P_s \tag{9}$$

On-host ticks exhibit density dependent survival. Studies demonstrate this may result from true density dependence or from density dependent host grooming behaviors as a result of tick exposure. To implement this, an exposure index ($EI$, Eq 10.) is calculated which measures the total number of ticks of each life stage ($L$, $N$, $A$; for larvae, nymphs, and adults, respectively) on each host type ($x$), scaled proportionally by mass of each life stage for the previous 8 weeks, with a loss of exposure of 0.44 per week. For each host type and tick life stage, there are given estimated minimum and maximum survival rates for high and low exposure rates. In between these bounds, on-host survival is determined by Eq 11, which yields a linear decrease in survival for increased $EI$.

$$EI = \sum_{i=1}^{9} 0.44^{i-1}(0.0021 L_{t-i} + 0.014 N_{(t-i)} + A_{t-i}) \tag{10}$$

$$S_o = \frac{S_{min} - S_{max}}{EI_{max} - EI_{min}}(EI - EI_{min}) + S_{max} \tag{11}$$

Ticks in the hardening stages are modeled with a constant survival, as molting success becomes the primary determinant of survival. Parameterization is based off of previous models to ensure realistic rates [26, 29].

## Infection

Transmission occurs during the on-host stages, starting in larvae, and passes upwards through life stages, but will not transmit from adult ticks to eggs [45]. Ticks disperse evenly between infected and susceptible hosts of the same species, and the rate of infection from hosts to ticks is determined by the competency of each host species, the proportion of infected hosts of each species, and the density of ticks per host. Competency for mice is set to 75%, and deer to 0%

[10, 17, 18, 24, 46]. The intermediate host serves as a generic species to maintain infection in the system in the absence of mice, and has a competency of 50%, in line with other small to medium sized mammal species [10, 17, 23].

Host infection is slightly more complicated. Infected ticks distribute evenly among hosts, but infected tick burdens may be in the range of only a few to no ticks per host throughout parts of the year. During these periods, it is expected that some hosts may have several ticks while others have none. We used a modification of a method established previously [26] which uses a Monte Carlo simulation to predict the rate of host infection, given the number of infected ticks per host ($ITH$) and assuming an infection rate of 100%, which is then scaled by the expected infection rate of ticks. This calculation is made several hundred times to ensure a robust calculation for infected tick burdens from 0.01 to 7.5 $ITH$, the upper range of which is sufficient to provide $> 99.9\%$ chance of infection. The $ITH$ is scaled by an infection rate of 0.9 to provide a 90% chance of infection per infected tick to host. For sufficiently high $ITH$ there will be an effectively 100% infection rate.

## Environmental-dependent variables and data

Average temperature and relative humidity are indicated as the most significant environmental factors which affect survival and development in *Ixodid* ticks [34–36, 47]. In the northeastern United States, where Lyme disease is most prevalent, humidity does not appear to play as significant a role, as humidity ranges do not typically fall outside of those optimal for tick survival [31]. As long term humidity data is less easily available, precipitation is sometimes used as a stand-in, which we have chosen to do. The average temperature is calculated as the average of the minimum and maximum temperature recordings for a day, and precipitation is calculated as an index, ($PI$), measured as $1/10$ *th* the current week's ($w + 1$) rainfall ($R$) in *mm* with a loss of 65% from the previous week's ($w$) index.

$$PI_{w+1} = R_{w+1} + 0.65PI_w \tag{12}$$

We obtained daily measurements of minimum and maximum temperature and precipitation from 22 sites throughout North America [48]. Data were selected to cover a range of locations and climate conditions from 1971–2021. Data were converted to weekly measurements and mean temperature and precipitation indices were calculated. Portland, Maine weather data was used as the primary location of focus for investigating model behavior.

## Simulation

All Simulations were run in Julia version 1.6.1 [49]. A variety of packages were used in the process of conducting simulations, analyzing data, and plotting in Julia [50–59] and in R [60–62]. To investigate the impact of host population dynamics on disease flow in the model, 10,000 simulations were run with mean and variance of the density distribution for all combinations of each variable ranging from 0–99 in increments of 1. Ranges were chosen to cover empirical measurements of density in the white-footed mouse, *Peromyscus leucopus*, and potential variation in these densities [21]. Each simulation was run with a time step of 52 weeks for 50 years. Simulations were repeated for each of the 22 chosen locations to investigate how changing environmental data affects model behavior. Simulation output consists of weekly time-series data for all tick and host stage classes within the model. A single simulation involves calculating the output of the discrete difference equations at each time step of one week, and using these result to calculate again for each successive time step until the completion of the desired number of weeks of simulation. The result is the time series data as described above, of which the last 10 years is stored as a column .csv file for each simulation, life stage, and host type.

## Model analysis

Analysis of model data was conducted in Julia version 1.6.1 [49] and additional packages [50–59], and plotting conducted with Julia and R version 4.0.5 [60–62]. To determine model prediction of disease risk, we calculated the mean, minimum, and maximum densities of infected and susceptible nymphs per year for the last 10 years of each simulation. These data were used to calculate standard disease risk measures. The density of infected nymphs, *DIN* was calculated as the mean, minimum, and maximum density of questing infected nymphs over the course of a year, with minimum and maximum density not necessarily aligned with the density of questing susceptible and infected nymphs. Density of nymphs, *DON* was calculated as the mean, minimum, and maximum density of the combined population of susceptible and infected nymphs. Nymphal infection prevalence, *NIP* was calculated as the mean, minimum, and maximum measures of the infected population divided by the combined population of susceptible and infected nymphs. Amplitude is measured as the yearly minimum subtracted from the maximum for these disease metrics. The mean over 10 years for each metric was used when analyzing results.

We completed a sensitivity analysis to determine the sensitivity of results to particular variables. Parameters were varied with a one at a time approach across a range from -10% to 10% of the baseline value. Sensitivity was calulcated as the percent change in the output measures (as are described above) divided by percent change in the parameter. Portland, Maine environmental data was used as well as a constant mouse density of 50 individuals per hectare.

## Results

### Effects of host dynamics on disease measures

We examined model data grouped by simulations run with no host dynamics (constant mouse densities) or the entire range of host dynamics ($\sigma^2$ = 1–99 mice) (S1 Table). *DIN*, Density of Infected Nymphs, and *DON*, Density of Nymphs, show similar patterns between dynamic and non-dynamic simulations, with the maximum and amplitude of each being much greater in simulations with host dynamics than those with no host dynamics. This pattern holds for effects on minimum and mean *DIN* and *DON*. *DIN* is more directly relevant to disease risk, so we will focus on results for *DIN* as a proxy for both. *NIP*, Nymphal Infection Prevalence, metrics show lowered averages and medians for mean, minimum, and maximum *NIP*, and increased amplitude between simulations with and without host dynamics. Minimum and maximum values remained constant for most *NIP* metrics. Minimum, maximum, mean, and amplitude in *DIN*, *DON*, and *NIP* were averaged over the final ten years of a simulation.

Mean *DIN* shows a positive linear relationship with mean mouse density, and a variable relationship with host dynamics ranging from a positive and slightly nonlinear relationship at low means to a negative and slightly non-linear relationship at high density (Fig 3A). Similar positive relationships with host dynamics and mean mouse density are demonstrated for amplitude, minimum, and maximum *DIN* (Figs 4A and 5A).

Measures for *NIP* show nonlinear relationships with both mean mouse density and host dynamics. These measures are positively related with mean mouse density (Figs 3B, 4B and 5B). With low host dynamics this relationship is a saturating curve, while with higher host dynamics this relationship becomes linear. Mean *NIP* decreases with increasing host dynamics but significant host dynamics compared to the mean mouse density begin to slow and eventually reverse this decrease, which is observable at low mean mouse densities (Fig 3B). The minimum point of this effect shifts to higher levels of host dynamics as the mean mouse density increases, with no minimum appearing in the range of explored host dynamics for higher

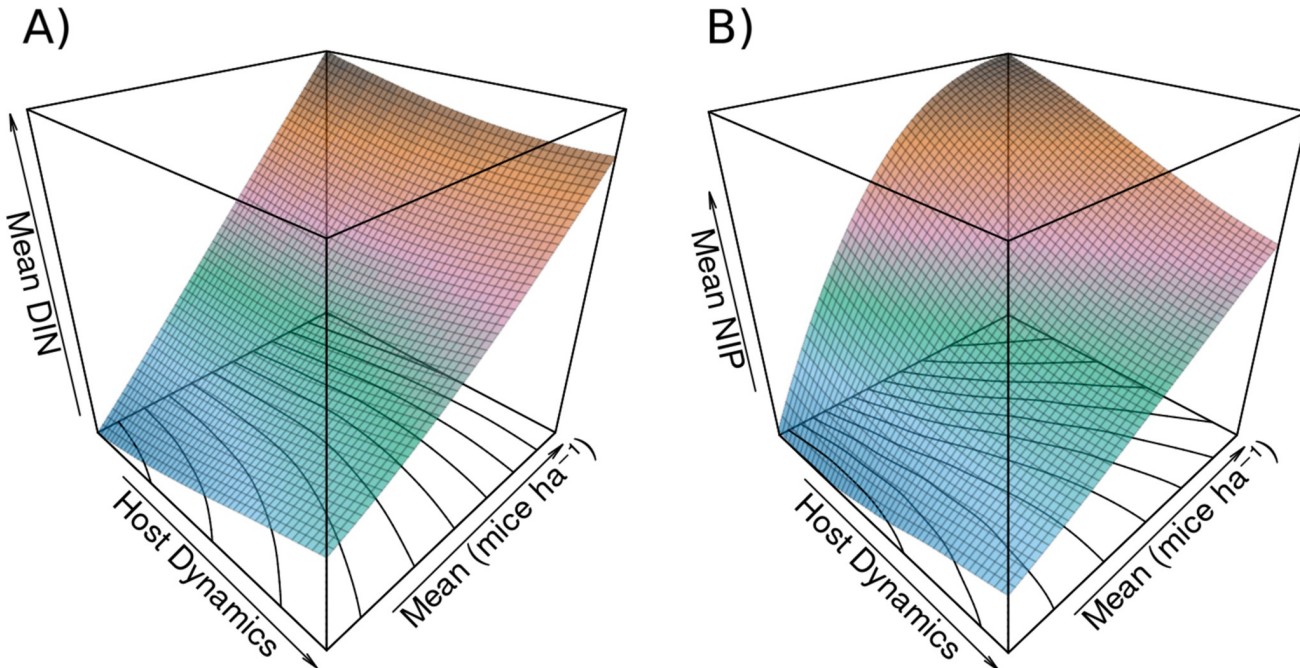

**Fig 3. Response of mean density of infected nymphs and nymphal infection prevalence to mouse population dynamics and mean mouse density.** A) shows the response of mean *DIN* to mouse population dynamics and mean density. The relationship to mean mouse density is positive and linear, while the relationship to host dynamics is weakly nonlinear, and shifts from a positive relationship to a negative relationship at low or high mean density, respectively. Contour lines plotted on the host dynamics × mean plane and coloring reveal surface features. B) shows the response of mean *NIP* to mouse population dynamics and mean density. The relationship between mean *NIP* and mean mouse density is a positive saturating curve under low population dynamics in the mouse population, which becomes a near linear relationship for high population dynamics. The relationship between host dynamics and *NIP* is negative, with higher host dynamics decreasing *NIP* at the given mean mouse density. At low mean densities, very high host dynamics begin to slow this decrease and even reverse it. Contour lines and coloration again reveal surface features.

mean densities. The amplitude of *NIP* follows a saturating curve across levels of host dynamics, reaching a maximum and slightly decreasing for high levels of host dynamics and high mean mouse density (Fig 4B). The minimum *NIP* shows a strong negative response to host dynamics (Fig 5B), while the maximum shows an unclear and relatively weak effect (S1 Fig), with a slight positive relationship at low mean mouse density shifting to a slightly negative relationship at high density.

The distributions of mean, minimum, and maximum *NIP* shows a negative shift in the median *NIP* measure with increasing host dynamics, with this shift being strongest in minimum *NIP* (S1 Table). Median amplitude in *NIP* shows a positive shift with increasing host dynamics (S1 Table). The median of mean and minimum *DIN* remains fairly stable, while amplitude and maximum *DIN* increase with host dynamics (S1 Table). The median of mean *DIN* remains nearly constant, while amplitude *DIN* increases, and the median of minimum and maximum *DIN* decrease and increase, respectively with increasing host dynamics (S1 Table).

## Effects of host dynamics under different climate conditions

We examined disease metrics across the selected locations to investigate patterns between sites and general geographic regions. *DIN* measures showed somewhat of an increase in mean, minimum, and maximum *DIN* under warmer environmental conditions, but showed no

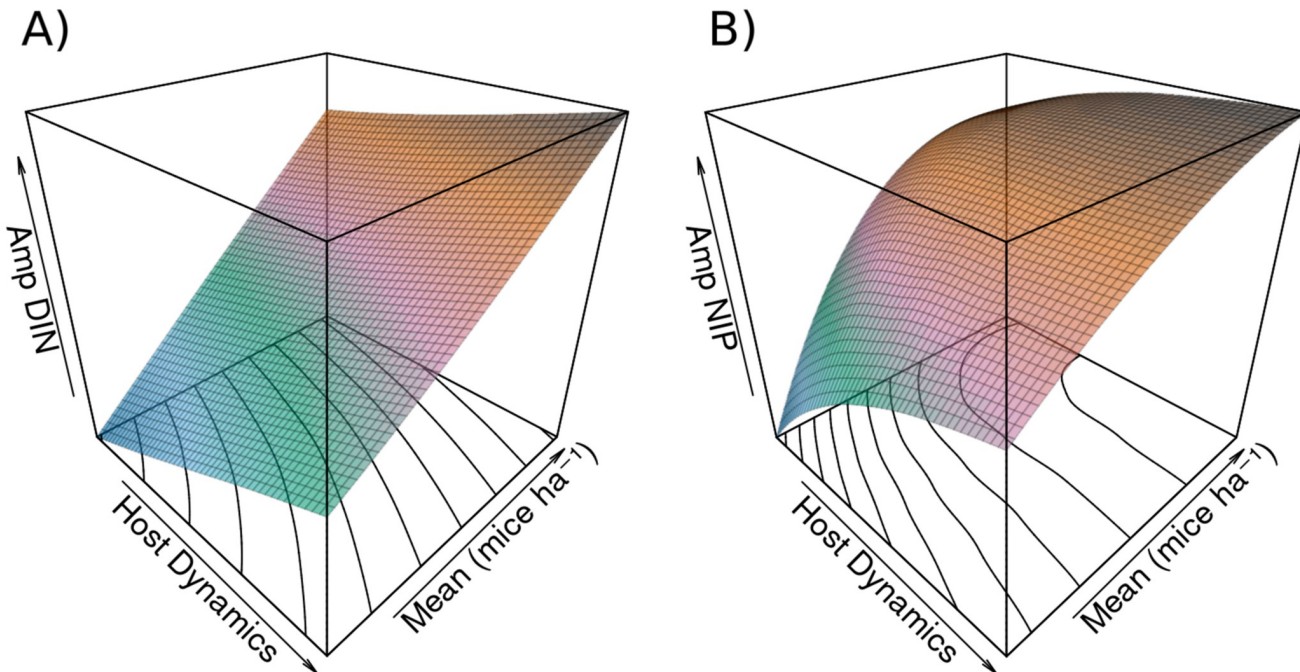

**Fig 4. Response of amplitude in density of infected nymphs and nymphal infection prevalence to mouse population dynamics and mean mouse density.** A) shows the response of amplitude in *DIN* to mouse population dynamics and mean density. The relationship to mean mouse density is positive and linear, while the relationship to host dynamics appears close to linear and is positive. Contour lines plotted on the host dynamics × mean plane and coloring reveal surface features. B) shows the response of amplitude in *NIP* to mouse population dynamics and mean density. The relationship between *NIP* and mean mouse density is a positive saturating curve under low population dynamics in the mouse population, which becomes linear as host dynamics increase. The relationship is a positive saturating curve between host dynamics and amplitude in *NIP*. At high mean densities the relationship to host dynamics is near linear, and shows signs of beginning to become negative at very high means and host dynamics. Contour lines and coloration again reveal surface features.

discernible changes in the pattern of this effect. Mean, minimum, and maximum *NIP* across locations shows very similar effects of mouse population dynamics with the range and inter-quartile range remaining similar while the median shifts down in response to host dynamics. This effect is apparent between geographic regions as well, though regions differ in their distributions for both *DIN* and *NIP*. Amplitude of *DIN* has a consistent pattern between locations and regions with or without host dynamics (Fig 6A). *NIP* amplitude has a markedly similar response to host dynamics across locations and regions despite variability with stable host populations (Fig 6B).

## Sensitivity analysis

Model outputs were largely robust to perturbations in parameter values (S2 Fig). Three parameters exceeded a sensitivity of 1%Δ output/%Δinput: $S_1$, the survival rate of mice (Sensitivity = −11.80); $\mu_{t,i}$ the optimal temperature for temperature-induced survival for immature ticks (Sensitivity = −2.16); and $\sigma_{t,si}$, the standard deviation of temperature-induced survival for immature ticks (Sensitivity = 1.68). Parameters that showed levels of sensitivity near but less than 1%Δ output/%Δinput exclusively relate to survival, host finding, and host burdens for immature ticks (larvae and nymphs). Several ecologically significant parameters, such as the lifespan of a tick lifestage and the fecundity of ticks, notably showed low sensitivities.

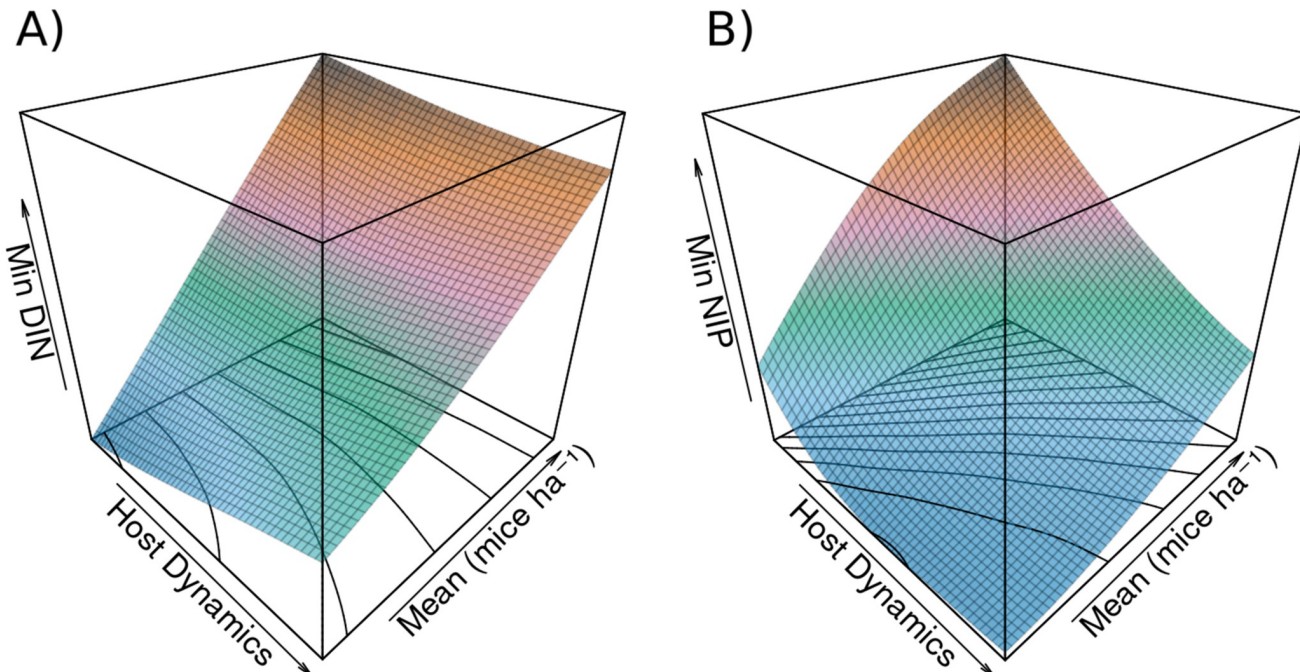

**Fig 5. Response of minimum density of infected nymphs and nymphal infection prevalence to mouse population dynamics and mean mouse density.**
A) shows the response of minimum *DIN* to mouse population dynamics and mean density. The relationship to mean mouse density is positive and linear, while the relationship to host dynamics appears weakly nonlinear, and shifts from a slight positive relationship at low mean density to a slight negative relationship at high mean density. Contour lines plotted on the host dynamics × mean plane and coloring reveal surface features. B) shows the response of minimum *NIP* to mouse population dynamics and mean density. There is a nonlinear positive relationship between minimum *NIP* and mean mouse density. The relationship between host dynamics and minimum *NIP* is negative, with higher host dynamics decreasing minimum *NIP* at the given mean mouse density. At low mean densities, very high host dynamics shows the slightest sign of beginning to increase the minimum *NIP*. Contour lines and coloration again reveal surface features.

## Discussion

Because (i) there is an observed positive correlation between host density, tick density, and *Borrelia burgdorferi* prevalence and (ii) variation tends to dampen dynamics, we expected Lyme disease risk metrics to loosely track host density with increasing host dynamics. In line with this prediction, our principle two findings were a negative effect of host dynamics on *B. burgdorferi* prevalence, and, curiously, we found no effect of host dynamics on tick density. We find the latter result particularly surprising and important, as it calls attention to an important facet of these complex ecological communities that will lead to a better understanding of and better predictions of *Borrelia burgdorferi* and Lyme disease risk dynamics. Below we speculate on our non-intuitive findings between host dynamics and disease risk, elaborate on how host dynamics affects disease risk temporally, how different and changing environmental conditions affect disease risk, and how we believe this study should be considered for future research.

### Associations between host dynamics and disease risk

Model results indicate that host dynamics in the mouse population have no effect on mean *DIN* and *DON*. While somewhat counterintuitive, this result is not as surprising as it may seem. With the high association between small mammal density and *DIN* [23, 24], host density is a very reliable predictor of tick density, which *DIN* and *DON* both measure. Tick density tracks host density, rising and falling in response to corresponding changes in the host

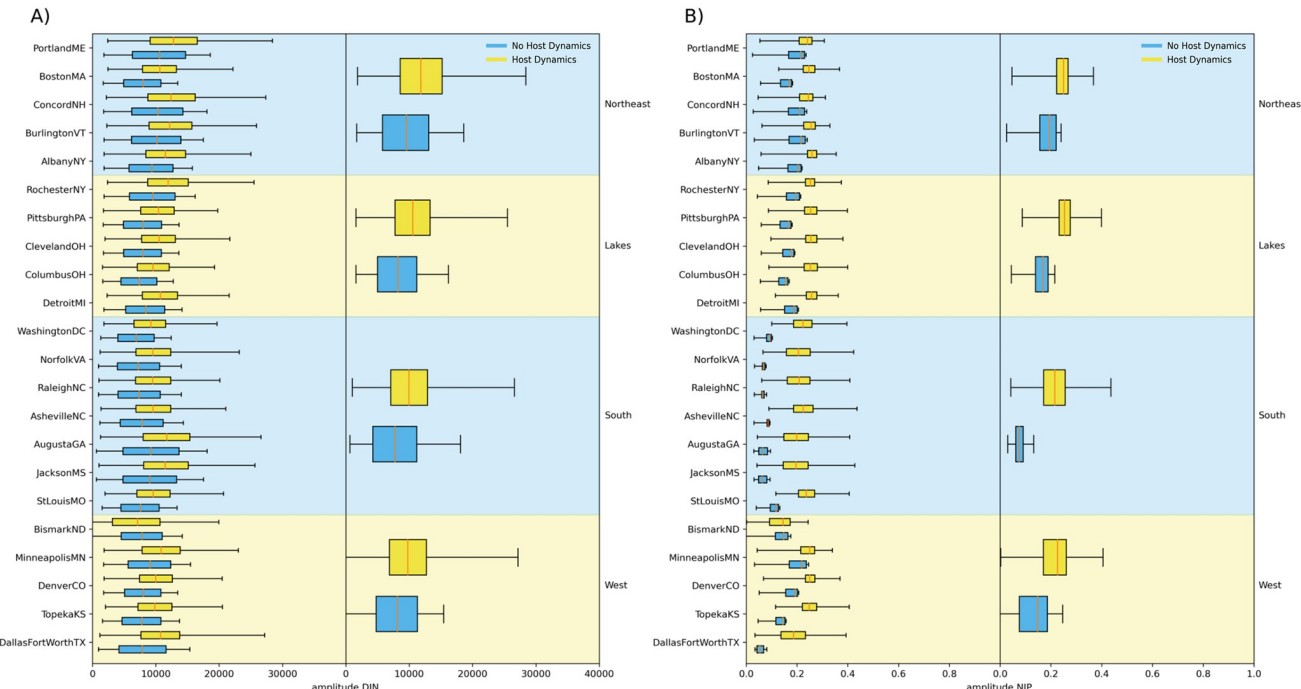

**Fig 6. Impact of mouse population dynamics on amplitude of density of infected nymphs and nymphal infection prevalence under varying environmental conditions.** A) displays the distributions of amplitude in *DIN* for locations with no host dynamics or with host dynamics for medium levels of host dynamics ($\sigma^2 = 33 - 67$). The left column shows boxplots for each location, while the right column shows boxplots for locations grouped by rough geographic area, with "West" indicating sites on the western range of Lyme disease and Eastern black-legged ticks. B) displays the same information for amplitude in *NIP*.

population. Tick density measures (*DIN*, *DON*) also track host density in the model with host dynamics, which averages out over multiple years of the simulation. Modeling studies have demonstrated that incorporating long-term empirical data on host density yields more accurate predictions of tick density when predicting disease risk at specific sites [26]. The effect on overall model output has not been determined, with research focusing on the qualitative accuracy of predictions on short timescales rather than multi-year simulations or investigation of model behavior. Studies that have considered host dynamics have focused solely on the effect of host density on disease risk rather than the effect of changing host density on disease dynamics. The results of our model still align with this research, and demonstrate the direct impact of host density on *DIN* and *DON*.

Long-term measurements of interannual variation in infection prevalence is lacking, and few modeling studies report yearly predicted *NIP* [26, 29, 30]. Available results show little variation in mean *NIP* with a constant host population, while simulations incorporating field data on host populations demonstrate lowered and more variable *NIP*, which aligns with our results. Empirical studies demonstrate this same variability in annual *NIP*, [24, 63, 64] which our results align with. Studies which examine variation in the host population also focus on the predictive ability of host density for *NIP*. The density of small mammal hosts has been found to be a poor predictor of infection prevalence [23, 24]. On the other hand, resources such as acorns can be a good predictor of mean *NIP*. This likely results from direct changes in the proportion of competent host species in the system, e.g. seed predators. The density of hosts does not necessarily indicate changing host proportions, and may instead result from

community-wide effects that change overall host density. In this model, we have demonstrated the scenario in which the proportion of competent hosts is changing by directly changing the density of a single host, and thus the proportional density, of hosts in the model system. The resulting effect on mean *NIP* is apparent in the results, and demonstrates the importance of considering relative host densities of competent and incompetent species rather than focusing on specific hosts or total host density in a system.

Our model shows that the strength of host dynamics is important, with a parabolic response of *NIP* to host dynamics apparent in model results (Fig 3B). This likely results because all mouse population means are simulated, and for each mean, the full range of host dynamics is also simulated. For mean host densities < 99 individuals ha$^{-1}$, the range of host dynamics will overlap with 0 individuals ha$^{-1}$ with high levels of host dynamics. Mouse density is bounded by 0 individuals ha$^{-1}$, as negative density of course makes no sense, so with greater host dynamics the true mean density of mice in a simulation will be higher than expected. Figs 3–5 could rather show the relation between true simulated mouse density, host dynamics, and *NIP*, but this neglects to show the nature of the response of *NIP* to host dynamics at a simulated mean. The magnitude of the variance of host dynamics must significantly exceed the mean mouse density before mean *NIP* begins to rise again (Fig 3B), showing that host dynamics have a strongly negative effect on *NIP* that remains even after mean host density has been boosted significantly. This observation indicates that the effect of host dynamics on infection prevalence is quite significant across different magnitudes and patterns of host dynamics.

## Temporal dynamics in disease risk

In the model, what we have defined as amplitude serves as an indicator of the range of a disease measure, and thus disease risk, over the course of a year. This quantifies how host dynamics and mean host density affects disease predictions in the model. A greater amplitude indicates greater contrast between least and most risky seasons, and a smaller range indicates similar risk between seasons. Understanding this range gives a sense of the pattern of disease as predicted by a model. The density of infected nymphs and total nymphs fluctuates seasonally, with both measures typically peaking in late spring or early summer to later in the year depending on how *DIN* is measured in a study. Changes in the amplitude of *DIN/DON* represent a change in the regular pattern of disease. The positive shift in amplitude for *DIN* shows a change in this pattern, and is a result of high host burdens supporting increased 'crops' of ticks in high host density years. The minimum unsurprisingly shifts very little, as this represents the sharp drop in tick densities as a result of mortality during periods of low activity and cold temperatures during the winter.

Intraannual variation data for *NIP* is very lacking, but the few studies that have been conducted indicate that seasonal *NIP* variability is indeed present in wild populations, and thus may present an important consideration when forecasting and managing disease risk through human behavioral changes [65, 66], and might also have value in seasonally focused wildlife management and disease prediction [67]. Mathematical models of Lyme disease risk, which often operate on sub-yearly time scales, predict this seasonality. This behavior is a feature of modeled interactions between populations of susceptible and infected ticks and hosts, which fluctuate throughout the course of a year in response to host births, deaths, and infection rates, which are related to periods of tick activity. Our model shows that host variation affects the amplitude of *NIP* in a year. The minimum range of *NIP* was most strongly affected, which suggests that host variation may be important in determining the magnitude of periods of low and high risk, likely in response to changes in the birth/death ratio throughout a year. Prior modeling studies have suggested that co-occurrence of host density increases and high questing tick

activity may boost tick density [67], but it is not known how this would affect *NIP*, and this was not explored in our model.

## Varying patterns in disease risk under a changing environment

While this model is not intended to be predictive of the specific dynamics of a particular area, qualitative changes in the response to host dynamics under different environmental conditions are of interest. We used the environmental conditions of different geographic locations to explore this idea. Model results show a latitudinal gradient in the magnitude of *DIN* and the pattern of the response of *NIP*. The increase in *DIN* measures at locations with higher annual temperatures is expected, as warmer winters result in greater tick survival and activity. The consistency of the response between locations for most measures is rather surprising, and suggests that similar communities of species will have similar disease dynamics at different locations, though the disease risk may be different between sites. The changing pattern in the effect of host dynamics on the amplitude of *NIP* at different locations, however, demonstrates the importance of considering host dynamics. It is apparent that environmental conditions may alter how disease risk interacts with host density and host dynamics at different sites. Incorporation of host dynamics should allow for greater understanding of the particular disease dynamics when forecasting disease risk at specific locations using mathematical models. Further study of how different kinds of host dynamics affects the prediction of disease risk will be valuable for better understanding our models of disease as well as patterns in disease under real world patterns of host dynamics.

## Further directions

Our results suggest that host dynamics, and more specifically the magnitude of variation in host population densities, is as an important consideration when modeling Lyme disease risk. More research is needed to understand how these dynamics may affect the use of theoretical models as exploratory and predictive tools. This might include simple to detailed modeling of host dynamics, or direct use of long-term population data. Predictive models should rely on either detailed modeling of host dynamics or, preferably, long-term host data to investigate location-specific disease dynamics. This will allow these models to more accurately describe disease at specific locations and will further the use of theoretical studies as investigative tools for the management of disease risk.

More research is needed to quantify important parameters impacting disease dynamics and our understanding of the ecology of ticks and tick-borne diseases. Further empirical studies should target key demographic parameters such as weekly survival and host-seeking rates. The relationships between environmental conditions and these rates also deserve further study and have especially important implications in the context of climate and environmental change. To understand how host populations influence disease risk, long-term studies can quantify host abundance, as well as the magnitude of variation in host populations. These data will aide our understanding of this complicated disease system, as well as further modeling efforts to guide management.

In our model, we assume instantaneous births at the beginning of a year, which will impact the ratio of infected to susceptible hosts when the host density is increasing. While this is a biologically unrealistic scenario, it presents a situation in which this shift in infected host prevalence drops as a result of reproduction, one of the important ways by which host dynamics are likely to impact disease risk, and especially the prevalence of infection. Any process that significantly shifts the proportion of susceptible hosts is likely to have repercussions on disease dynamics. This might occur in the real world as a result of normal population dynamics such

as overwintering deaths in host species, seasonal periods of reproduction, or as a result of increased resource levels. In future models, host population dynamics might be modeled on a shorter timescale, reflecting seasonal birth and death rates or the response to resource levels such as seed masting.

An important point to investigate is how the density of hosts in one year impacts disease measures in successive years in a host dynamics simulation. This can be explored using current model data to investigate how the relationship between host density and disease changes at different levels of host dynamics. It will also be important to determine how the timing of host population changes might affect the timing of peaks in disease risk. It has been shown that this can affect tick density [67], but the interactions of patterns in host density with *DIN* and *NIP* have not been explored. This can be examined in our model by changing the week in which the new mouse population is chosen or by introducing a more complex model of mouse population dynamics. As has been suggested with *DON* and the interaction between host density and tick density peaks, the timing of tick activity, questing, and life stage peaks is likely to factor into further layers by which hosts may influence disease dynamics.

This project has demonstrated the importance of considering population dynamics in tick hosts when modeling Lyme disease risk. This model only begins to touch on potential outcomes on model behavior with the incorporation of very simple host dynamics. Potential avenues of expansion and further exploration with this model are many, and they offer strong potential to further our understanding of Lyme disease risk. With a complete understanding of how host, tick, and disease dynamics interact, we can begin to understand when and when not to emphasize different pieces of this complex system. Further exploration of the impact of host dynamics on disease risk will hopefully increase our knowledge of how Lyme disease spreads and behaves, and aide the development of models which are able to more accurately study and predict disease risk.

## Supporting information

**S1 Fig. Response of maximum density of infected nymphs and nymphal infection prevalence to variance in the mouse population and mean mouse density.** A) shows the response of maximum *DIN* to mouse variation and mean density. The relationship to mean mouse density is positive and linear, as is the relationship to mouse variance. Contour lines plotted on the variance × mean plane and coloring reveal surface features. B) shows the response of maximum *NIP* to mouse variation and mean density. There is a nonlinear positive relationship between maximum *NIP* and mean mouse density. The relationship between mouse population variance and maximum *NIP* is inconsistent, with higher variance increasing maximum *NIP* at the low mean mouse density. At higher mean densities, maximum *NIP* transitions to decreasing with variance. Contour lines and coloration again reveal surface features.
(TIF)

**S2 Fig. Sensitivity analysis.** This figure depicts the results of the sensitivity analysis we conducted. Parameters changed appear on the y-axis, with each point on the x-axis indicating the sensitivity measures (percent change in output/percent change in parameter). Red dots indicate the mean sensitivity of each parameter.
(TIF)

**S1 Table. Summary statistics, grouped by x, DIN, NIP.** This table presents summary statistics for disease risk metrics, grouped by different levels of mouse population variance.
(PDF)

**S2 Table. Tick life stages.** Tick life and developmental stages represented in the model.
(PDF)

**S3 Table. Demographic tick processes.** Tick demographic processes represented in the model.
(PDF)

**S4 Table. Parameters used in the model.**
(PDF)

**S5 Table. Parameters used in the model, continued.**
(PDF)

**S1 File. Model equations.** The equations representing tick, host, and infection processes in the model are presented here. See S4 and S5 Tables for parameter names and purposes.
(PDF)

## Acknowledgments

We would like to express gratitude to Dr. Allison K. Barner and Dr. Stephanie R. Taylor for their input and guidance on this project and previous manuscript versions.

## Author Contributions

**Conceptualization:** Joseph D. T. Savage, Christopher M. Moore.

**Data curation:** Joseph D. T. Savage.

**Formal analysis:** Joseph D. T. Savage.

**Investigation:** Joseph D. T. Savage.

**Methodology:** Joseph D. T. Savage.

**Supervision:** Christopher M. Moore.

**Visualization:** Joseph D. T. Savage, Christopher M. Moore.

**Writing – original draft:** Joseph D. T. Savage.

**Writing – review & editing:** Joseph D. T. Savage, Christopher M. Moore.

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
