## [Decision Letter · Decision Letter 0]

19 Sep 2023

PONE-D-23-23254How do host population dynamics impact Lyme disease risk dynamics in theoretical models?PLOS ONE

Dear Dr. Savage,

Thank you for submitting your manuscript to PLOS ONE. After careful consideration, we feel that it has merit but does not fully meet PLOS ONE’s publication criteria as it currently stands. Therefore, we invite you to submit a revised version of the manuscript that addresses the points raised during the review process.

We look forward to receiving your revised manuscript.

Kind regards,

Catherine A. Brissette, Ph.D.

Academic Editor

PLOS ONE

Journal Requirements:

Additional Editor Comments:

I apologize for the delay, it has been difficult to find reviewers. Please address the concerns raised by reviewer #1, in particular, clarifying your research question.

Reviewers' comments:

Reviewer's Responses to Questions

**Comments to the Author**

1. Is the manuscript technically sound, and do the data support the conclusions?

Reviewer #1: Yes

2. Has the statistical analysis been performed appropriately and rigorously? 

Reviewer #1: Yes

3. Have the authors made all data underlying the findings in their manuscript fully available?

Reviewer #1: Yes

4. Is the manuscript presented in an intelligible fashion and written in standard English?

Reviewer #1: Yes

5. Review Comments to the Author

Reviewer #1: The models seem to be well constructed and parameterization is based on realistic estimates of many of the variables. The authors explore several determinants of environmental and host impacts on tick-based disease metrics such as DON, DIN, and NIP. Some conceptual structure would help place the results in a ecological context. For instance section “3.1 Site-specific effects of variance” could specify what kind of variance is being tested. In this case, it is mouse density. But the subheading suggests that it is evaluating site level differences. In general, the manuscript would be greatly improved by developing clear research questions and hypotheses. As currently written, it is unclear what the research questions are. The authors develop a interesting dynamic, three-host tick-borne disease model that could be useful to the field to address key questions on the importance of host and environmental variables on risk of Lyme disease but there needs to be more biological context for the models presented.

Formatting is a bit odd with figure and table legends embedded throughout the text, but without the actual table or figure. It is a little hard to tell when the legends end and the regular text begins. I’m not sure if this is a formatting requirement by the journal or something the authors chose to do but I had a difficult time understanding the main flow of the text because of the legends interspersed throughout.

There are also several locations in the manuscript where the is missing information, as indicated by “???”. For example Lines 217, 403, 425. This manuscript lacks polish.

The finding that tick density tracks host density is interesting. Was there a lab in the model? Given that nymphs are active the year after larval ticks feed on mice? Then later on lines 222, the authors seem to contradict their finding that density of hosts is a poor predictor of disease risk based on other studies 17, 20. How do the authors reconcile their results with these other results?

Some of the discussion (e.g. Lines 239-246) could be framed more generally and read more like results like discussion.

Line 85: I think you mean the “flow of the pathogen”

Methods

Line 235: I don’t think ha needs to be squared as it is an area already.

Line 371: is infection clearing referring to the tick or the hosts?

Line 374: I don’t think intermediate host is appropriate here as it often refers to complex parasite life cycles. Probably just “intermediate SIZED host” would be more clear.

Line 403 and 425: “Equation ??” refers to which one?

6. PLOS authors have the option to publish the peer review history of their article (what does this mean?). If published, this will include your full peer review and any attached files.

Reviewer #1: No

---

## [Author Response · Author response to Decision Letter 0]

30 Nov 2023

Thank you to the editorial board at PLOS ONE and to our reviewer for taking the time to review and comment on this manuscript. We have addressed the points raised by the editor and reviewer. Below are our responses to the raised points. 

 Editorial points:

 1. We have ensured that our mansucript meet's PLOS ONE's style requirements, and made adjustments to fit these requirements as needed. 

 2. We have removed funding information from the acknowledgements. 

 3. We acknowledge our data availability statement, and will be maintaining it. We will provide relevant information for publicly accessing data prior to publication. 

 4. Note an update to our financial disclosure in the cover letter. 

 Reviewer Points:

 1. "Conceptual structure would help place the results in a ecological context."

 We have addressed this point by taking the suggested example, and reviewing the manuscript to improve the ecological context of our results. Adjustments have been made to the structure and language of the manuscript to further this goal. 

 2. "In general, the manuscript would be greatly improved by developing clear research questions and hypotheses"; and 3. The manuscript needs more biological context

 We have addressed these points by restructuring the introduction to the manuscript, adding clearer language, and improving the biological context of our research goals throughout the manuscript. We believe that the manuscript now better reflects the significance of our questions and how we aimed to investigate them. 

 4. Formatting.

 We have verified that our formatting follows PLOS ONE's submission standards, apologies for any difficulties in legibility. 

 5. Missing detail in the manuscript/lack of polish. 

 The errors indicated by our reviewer have been corrected, and we have made further edits to the manuscript to add polish and important details. 

 6. "The finding that tick density tracks host density is interesting....Then later on lines 222, the authors seem to contradict their finding that density of hosts is a poor predictor of disease risk based on other studies 17, 20. How do the authors reconcile their results with these other results?"

 We have clarified this section in the manuscript, removing some ambiguous language that could have led to this conclusion. To clarify briefly, the density of hosts has been found to be a poor predictor of infection prevalence, one measure of disease risk. However, the density of hosts is a good predictor of the density of ticks and infected ticks. The previous draft did not make clear which measure of disease risk was being referred to, and this has been corrected. There are lags in the model, but we have focused on a broader look at disease risk measures predicted by the model with or without host dynamics. 

 7. Suggested edits and typos. 

 We adpoted corrections for the typos and clarifications suggested by the reviewer, and edited the manuscript further to increase clarity and remove typos. 

In addition to these edits, we have reviewed the manuscript and made general edits to improve the clarity of the manuscript, and address concerns on the polish and biological context of the manuscript and research goals. We are grateful for the opportunity to submit this manuscript for further review.

---

## [Decision Letter · Decision Letter 1]

23 Jan 2024

PONE-D-23-23254R1How do host population dynamics impact Lyme disease risk dynamics in theoretical models?PLOS ONE

Dear Dr. Savage,

Thank you for submitting your manuscript to PLOS ONE. After careful consideration, we feel that it has merit but does not fully meet PLOS ONE’s publication criteria as it currently stands. Therefore, we invite you to submit a revised version of the manuscript that addresses the points raised during the review process.

I know this decision is frustrating, but you have received two very disparate reviews. However, I feel that you responded to the previous critiques, and I disagree with Reviewer 2's outright dismissal of the manuscript. Please provide clarity on the equations in particular as indicated by Reviewer 2, and address the minor comments by Reviewer 3. You can disregard  the comments from Reviewer 2 regarding organization of the manuscript, as long as the submission followed PLOS guidelines.==============================

We look forward to receiving your revised manuscript.

Kind regards,

Catherine A. Brissette, Ph.D.

Academic Editor

PLOS ONE

Reviewers' comments:

Reviewer's Responses to Questions

**Comments to the Author**

1. If the authors have adequately addressed your comments raised in a previous round of review and you feel that this manuscript is now acceptable for publication, you may indicate that here to bypass the “Comments to the Author” section, enter your conflict of interest statement in the “Confidential to Editor” section, and submit your "Accept" recommendation.

Reviewer #2: (No Response)

Reviewer #3: (No Response)

2. Is the manuscript technically sound, and do the data support the conclusions?

Reviewer #2: No

Reviewer #3: Yes

3. Has the statistical analysis been performed appropriately and rigorously? 

Reviewer #2: No

Reviewer #3: Yes

4. Have the authors made all data underlying the findings in their manuscript fully available?

Reviewer #2: Yes

Reviewer #3: Yes

5. Is the manuscript presented in an intelligible fashion and written in standard English?

Reviewer #2: No

Reviewer #3: Yes

6. Review Comments to the Author

Reviewer #2: How do host population dynamics impact Lyme disease risk dynamics in theoretical models?

Summary: Lyme disease is caused by Borrelia burgdorferi and this spirochete bacterium is transmitted among vertebrate hosts by black-legged ticks (Ixodes scapularis). Larval ticks feed on rodent reservoir hosts, and moult into nymphs. Nymphs feed on the same set of rodent reservoir hosts the following year and moult into adult ticks. Adult ticks feed on larger vertebrate hosts like deer (they do not feed on deer). The density of infected nymphs (DIN) is the most important predictor of the risk of Lyme disease although numerous studies also present the nymphal infection prevalence (NIP). The rodent population size can fluctuate dramatically from year-to-year, which drives variation in larval feeding success and in the abundance of nymphs and adult ticks in the following years (Bregnard et al., 2020; Ostfeld et al., 2006).

The purpose of the present study is to investigate how fluctuations in the rodent population size will influence measures of Lyme disease risk such as the DIN and NIP. The authors create an ecological model that keeps track of 4 tick stages (eggs, larvae, and nymphs, and adult ticks) and 3 different vertebrate hosts, rodents, a medium mammal species, and deer. The population densities of the medium mammal species and deer are kept constant. The population density of the rodent population is allowed to fluctuate. Figure 6 compares the situation with constant mouse density (left panel) versus fluctuating mouse density (right panel). The main results are shown in Figures 1, 2, and 3, which demonstrate the effect of mean mouse density and host dynamics on the mean, amplitude, and minimum DIN and NIP. In the methods, I could not find what the authors meant with ‘dynamics’, but I think that it refers to the variance in the abundance of mouse density. For Figure 1, the authors show that the mean DIN and mean NIP increase with mouse density and this result is expected because more mice feed more ticks and because mice are the most competent reservoir host for B. burgdorferi. Host dynamics (again, I think that this means variance in mouse population size) has a slightly negative effect on the mean DIN and NIP. The results for the minimum DIN and NIP (Figure 3) largely mirror the results for the mean DIN and NIP (Figure 1). For Figure 2, the response variable is the amplitude in the DIN and NIP, which refers to the difference between the yearly maximum and the yearly minimum over a 10-year period (I think, it’s not totally clear). Figure 2 shows that as you increase the mean mouse density and the variance in mouse density, the variance (amplitude) in the DIN and NIP increases. This result is expected. The authors have constructed a model where mouse biomass is converted to tick biomass. If we increase mouse biomass and the variance in mouse biomass, the result is inevitable that the mean and variance of tick biomass will increase. I am not sure that a model was needed to show this very basic and expected result.

Concerns about biological reality of the model

I have concerns about the biological reality of this model. The authors calculate environmentally dependent survival of ticks by multiplying temperature by the precipitation index. How is this valid? The authors do not provide any justification as to how the product of two climate variables estimates the life history trait of an organism. In this model, adult ticks feed on rodent reservoir hosts, but this is not the case in nature. It would be reassuring to know whether the tick population dynamics resemble what happens in nature. Unfortunately, Figure 6 shows the population dynamics for larvae and eggs, but not for the nymphs and adult ticks. In year 3 (weeks 104 to 156) there is a peak in mouse density and the larval ticks have high feeding success. As a result, the abundance of nymphs in year 4 (weeks 156 to 208) should be high, but we cannot evaluate this because the line for the nymphs is not shown. The models show that the mean DIN and NIP increase as mouse density increases (Figure 1), and this is entirely expected because mice feed ticks and because mice have the highest reservoir competence for B. burgdorferi (LoGiudice et al., 2003).

Comments on organization of manuscript

This manuscript was difficult and frustrating to review because it was poorly organized. Figures are much easier to review if they are pasted into the Word document with the legend immediately below. In every manuscript that I have ever reviewed, the figure legends are at least grouped together at the end of the manuscript. In the present manuscript, they were inserted as paragraphs in the text of the manuscript, which made them difficult to find. The figures were not listed in order at the end of the manuscript (Figures 4, 5, 6, 1, 2, 3).

Comments on the figures

The Y-axes in Figures 1, 2, and 3, should have real units for the DIN and NIP. A reasonable model should generate realistic values for the DIN and NIP. Because Figures 1, 2, and 3 do not include units for the DIN and NIP, we cannot evaluate whether the model is producing realistic variables. The figure quality for figure 4 is so poor that I could not read any of the text. In Figure 6, the color scheme is incredibly confusing: blue indicates eggs in the top row, larvae in the middle row, and mice in the bottom row; yellow indicates larvae in the top row but nymphs in the middle row; green indicates nymphs in the top row and adults in the middle row. For the top row, the nymphs and adult ticks are indicated in the legend, but they are not show in the graphs.

Confusing presentation of equations

In most modelling articles that I have reviewed, the authors usually provide a table that lists all of the parameters that are used in the model (Ogden et al., 2007; Ogden et al., 2005; Ogden and Tsao, 2009). In most modelling articles that I have reviewed, the rationale for the equation is described in the text and the equation is cited in the text. In the present manuscript, the authors define some of the variables in the text and then equations suddenly appear without explanation. I give more detailed examples below.

Section 5.2

In section 5.2 (lines 399-404), the authors define the parameters M, S, I, and ∆M. Then the legend for Figure 6 appears (very confusing). Then equations 1, 2, 3, and 4 appear without any explanation. Being familiar with models, I can figure out that the subscripts of y and y + 1 refer to different years, but this is not explained in the manuscript.

Section 5.3

In section 5.3 (lines 428-430), the authors cite equation 11, but they are referring to equation 5. Again, they don’t cite equations 5 and 6 in the text. In the text (line 428), they present a parameter a(i,x) without its subscripts. They present that subscript i refers to the tick stage but don’t present that the subscript x refers to the host. This is confusing.

Section 5.4

In section 5.4 (line 450), the authors do not cite equations 8, 9, 10 and 11. They do cite equation 5, when they mean to cite equation 11. The parameters L(t-i), N(t-i), A(t-i) in equation 10 are not presented in the main text. The parameters So, Smin, and Smax in equation 11 are not explicitly defined in the text, but they appear to refer to survival. I believe that environmentally dependent survival (Se) in equation 9 occurs is related to the survival parameters in equations 10 and 11, but the relationship is not clear.

Equation 4 appears to be wrong

Equation 4 shows two equations, and I believe that the second one is wrong and should read, I(y+1) = Sy + ∆M [Iy/Sy + Iy]. Otherwise, the two equations for S(y+1) and I(y+1) are identical, which does not make sense.

Biological basis of equation 9 is not clear

Survival of ticks in the environment can depend on abiotic variables like temperature and survival. Previous population ecology models of Ixodes scapularis by Ogden and colleagues modelled developmental transitions (e.g., engorged larva molting into nymphs) as a function of temperature (Ogden et al., 2007; Ogden et al., 2005; Ogden and Tsao, 2009). At low temperatures, the time delays become so long that the ticks never go through the development transition. In equation 9 of the present study, the authors calculate environmentally dependent survival of ticks (Se) as the product of temperature (Ts) and the precipitation index (Ps). How can you multiply two abiotic climate variables together and get an estimate of tick survival? Survival is a proportion measured over a unit time. The authors do not specify the units of temperature and the precipitation index, but let’s imagine that they are measured in degrees Fahrenheit and inches of precipitation per year. The State of Maine has a mean annual temperature of 25 degrees Fahrenheit and an annual precipitation of 40 inches. The product of these two variables is 1000 degrees Fahrenheit * inches of precipitation, which the authors claim is equal to the temperature-dependent survival. This does not make any sense. You can’t multiply two abiotic climate variables and expect that to equal tick survival.

References

Bregnard, C., Rais, O., Voordouw, M.J., 2020. Climate and tree seed production predict the abundance of the European Lyme disease vector over a 15-year period. Parasit Vectors 13(1), 408. https://doi.org/10.1186/s13071-020-04291-z.

LoGiudice, K., Ostfeld, R.S., Schmidt, K.A., Keesing, F., 2003. The ecology of infectious disease: Effects of host diversity and community composition on Lyme disease risk. Proc Natl Acad Sci U S A 100(2), 567-571.

Ogden, N.H., Bigras-Poulin, M., O'Callaghan, C.J., Barker, I.K., Kurtenbach, K., Lindsay, L.R., Charron, D., 2007. Vector seasonality, host infection dynamics and fitness of pathogens transmitted by the tick Ixodes scapularis. Parasitology 134, 209-227.

Ogden, N.H., Bigras-Poulin, M., O'Callaghan, C.J., Barker, I.K., Lindsay, L.R., Maarouf, A., Smoyer-Tomic, K.E., Waltner-Toews, D., Charron, D., 2005. A dynamic population model to investigate effects of climate on geographic range and seasonality of the tick Ixodes scapularis. Int J Parasitol 35, 375-389.

Ogden, N.H., Tsao, J.I., 2009. Biodiversity and Lyme disease: Dilution or amplification? Epidemics 1(3), 196-206. https://doi.org/10.1016/j.epidem.2009.06.002.

Ostfeld, R.S., Canham, C.D., Oggenfuss, K., Winchcombe, R.J., Keesing, F., 2006. Climate, deer, rodents, and acorns as determinants of variation in Lyme-disease risk. PLOS Biol 4(6), 1058-1068.

Reviewer #3: In this research article, the authors describe a mathematical model to investigate the impact of host dynamics on vector-associated Lyme disease risk metrics. Using a simplified model of the complex Lyme disease ecological cycle, results show that altering host dynamics of key small mammal reservoir hosts has relatively little effect of overall tick infection dynamics. This has implications for future Lyme ecology modelling studies that are used to estimate Lyme disease risk.

Whilst a necessary over-simplification of the Lyme disease enzootic cycle, the model constructed is relevant and quite impressive, with different tick life stages, survival, on and off host populations, and infected/uninfected populations represented among multiple other parameters. The methods section gives a good description of the model that is easily understandable to non-modellers. This will be of interest to researchers in the disease modelling and Lyme ecology fields. Further modifications of this model utilising different host populations etc. will be useful in future for investigating effects of different host population parameters on Lyme ecology.

I recommend a few minor changes to improve the manuscript:

Figure 6 - nymphs and adults are not visible in parts (a) and (d). It woulfd be useful to be able to see the seasonal peaks of these life stages, so perhaps they could be shown by increasing numbers to 2x nymphs and 20x adults, as in parts (b) and (f).

Discussion

Be careful with the wording of Lyme vs. Borrelia. Lyme refers to the human disease, whilst this study is rather predicting Lyme disease risk based on tick infection prevalence.

line 212 and 215 - change 'Lyme prevalence' to 'nymph infection prevalence' or 'Borrelia prevalence'.

line 218 - similarly change Lyme dynamics to Borrelia or nymph infection dynamics

line 322 - this would be accurate if it said 'modeling Lyme disease risk'

7. PLOS authors have the option to publish the peer review history of their article (what does this mean?). If published, this will include your full peer review and any attached files.

Reviewer #2: No

Reviewer #3: No

---

## [Author Response · Author response to Decision Letter 1]

4 Apr 2024

Thank you to the editorial board at PLOS ONE and to the reviewers for taking the time to review and comment on this manuscript. We have addressed the points raised by the reviewers. Below are our responses to the raised points. 

Reviewer #2

1. Clarity on what host dynamics means

 - We have updated the introduction to better highlight the meaning of host dynamics in our study, with simulated variation around a mean host density representing a dynamic host population. 

2. Clarity on disease metrics. 

 - We have added language to the results section to better clarify the disease metrics, which are also described in greater detail in the methods. 

3. Concerns of biological realism. 

 - "The authors calculate environmentally dependent survival of ticks by multiplying temperature by the precipitation index."

 - We have utilized standard techniques in calculating an environmentally dependent survival rate. Both temperature and humidity (for which the precipitation index is a proxy) have independent relationships with survival. This relationship reduces survival proportionate to how close the environmental conditions are to the optimal values (e.g. temperature, or precipitation index). As temperature and precipitation index both have this effect, the reduction can be represented multiplicatively in the model equations. We appreciate the recomended papers to review, but note that these models only consider one environmental variable (temperature). See for example Gaff et al. 2020 for the method we utilized for considering the impact of two environmental conditions on survival. 

 - for possible confusion surrounding this equation, it is important to note that we are not multiplying environmental variables, but functional responses of survial to these variables. 

4. Adults feed on rodent hosts.

 - This is not the case and may have been a misreading of the text.

5. Figure 6 clarity.

 - We have made the suggestions to the color scheme and axes of this figure. 

6. Figure 1: Mean Nymphal Infection Prevalence (NIP) decreases with increase host dynamics.

 - This is not the case. We do understand the implication that if this were the case, the results would be quite expected, and diminsh the value of the paper. However, increasing host dynamics actually have a negative impact on mean NIP. 

7. Figure 2 & 3: basic/expected results.

 - These figures demonstrate that, while amplitude follows expected patterns for both DON and NIP, it is in the minimum NIP measure where we see a strongly negative nonlinear effect of increasing host dynamics. We believe that these figures are important in helping explain and interpret the finding that mean NIP decreases with increasing host dynamics. 

8. Axis labels on figures 1-3. 

 - These figures are intended to be qualitative in nature. As they represent 3D surfaces of smoothed model output and cannot be rotated interactively, axis labels would do little to show features on these surfaces. Note that quantitative information for these figures are included in S1 table, and cited in the text where approrpiate. 

9. Manuscript organization. 

 - We have followed PLOS guidelines for the manuscript, and believe that we are in accordance with those guidelines to the best of our abilities. 

 - Figure 4 we have checked to ensure good resolution, and believe the resolution error is a feature of the compiler. We will confirm that the figure is legible on resubmission. 

10. Confusing representation of equations.

 - We have confirmed that all parameters in the equations included in the manuscript are defined in the text preceeding them, and have corrected the noted exceptions. We have also included a reference to the parameter tables S4 and S5 in the text for interpreting the equations in S1 file. 

 - we have ammended the noted typo in the equations, and proofed for further typos. 

11. References to equations are incorrect. 

 - We have ammended these typos and ensured that equations are referenced in the text. 

Reviewer #3.

1. Figure 6 improvements

 -We have made the suggested edits to figure 6. 

2. Clear language on Lyme Disease vs. Pathogen prevalence

 - We have proofed further to ensure that we are not conflating Lyme disease with the pathogen. 

We have made further edits where clarity felt useful based on reviewer feedback, and are grateful for this additional opportunity for revision.

---

## [Editor Report · Decision Letter 2]

16 Apr 2024

How do host population dynamics impact Lyme disease risk dynamics in theoretical models?

PONE-D-23-23254R2

Dear Dr. Savage,

We’re pleased to inform you that your manuscript has been judged scientifically suitable for publication and will be formally accepted for publication once it meets all outstanding technical requirements.

Kind regards,

Catherine A. Brissette, Ph.D.

Academic Editor

PLOS ONE
---

## [Editor Report · Acceptance letter]

26 Apr 2024

PONE-D-23-23254R2 

PLOS ONE

Dear Dr. Savage, 

I'm pleased to inform you that your manuscript has been deemed suitable for publication in PLOS ONE. Congratulations! Your manuscript is now being handed over to our production team.

Kind regards, 

on behalf of

Dr. Catherine A. Brissette 

Academic Editor

PLOS ONE